# Reoccurring neural stem cell divisions in the adult zebrafish telencephalon are sufficient for the emergence of aggregated spatiotemporal patterns

Valerio Lupperger [1], Carsten Marr [1]*, Prisca Chapouton [2,3‡]*

**1** Helmholtz Zentrum München–German Research Center for Environmental Health, Institute of Computational Biology, Neuherberg, Germany, **2** Helmholtz Zentrum München–German Research Center for Environmental Health, Unit Sensory Biology & Organogenesis, Neuherberg, Germany, **3** Helmholtz Zentrum München–German Research Center for Environmental Health, Institute of Stem Cell Research, Biomedical Center, Faculty of Medicine, LMU Munich, Planegg, Germany

‡ Lead contact

* carsten.marr@helmholtz-muenchen.de (CM); chapouton@helmholtz-muenchen.de (PC)

**Data Availability Statement:** We provide access to raw image data, processed single cell information, code for analysis, parameter inference and result

## Abstract

Regulation of quiescence and cell cycle entry is pivotal for the maintenance of stem cell populations. Regulatory mechanisms, however, are poorly understood. In particular, it is unclear how the activity of single stem cells is coordinated within the population or if cells divide in a purely random fashion. We addressed this issue by analyzing division events in an adult neural stem cell (NSC) population of the zebrafish telencephalon. Spatial statistics and mathematical modeling of over 80,000 NSCs in 36 brain hemispheres revealed weakly aggregated, nonrandom division patterns in space and time. Analyzing divisions at 2 time points allowed us to infer cell cycle and S-phase lengths computationally. Interestingly, we observed rapid cell cycle reentries in roughly 15% of newly born NSCs. In agent-based simulations of NSC populations, this redividing activity sufficed to induce aggregated spatiotemporal division patterns that matched the ones observed experimentally. In contrast, omitting redivisions leads to a random spatiotemporal distribution of dividing cells. Spatiotemporal aggregation of dividing stem cells can thus emerge solely from the cells' history.

## Introduction

Somatic stem and progenitor cells, basic units of tissue maintenance and growth, can be found in distinct states, either dividing or quiescent. The duration of the quiescence state is not predictable [1], and its regulation has a profound impact on healthy tissue maintenance. Therefore, understanding the mechanisms of adult stem cell cycle regulation is crucial. Stem cells, when compartmentalized in areas where neighboring cells fulfill structural and niche functions, can be regulated locally, such as the hair follicle in the skin [2]. However, in a homogeneous population of equipotent stem cells that reside under the same conditions, it is unclear what drives distinct behaviors of quiescence or cell cycle entry. This raises the question of how

figures at https://github.com/QSCD/spatiotemporalAnalyses. Raw image data can be accessed from https://hmgubox2.helmholtz-muenchen.de/index.php/s/MBrLMra7wpj2qKn.

**Funding:** V.L. was funded by the Bundesministerium für Bildung und Forschung (BMBF, https://www.gesundheitsforschung-bmbf.de/de/micmode-i2t-modulare-bildanalyseplattform-zur-integration-von-mikroskopischen-bildbasierten-7611.php), grant 01ZX1710A-F (Micmode-I2T). C.M. has received funding from the European Research Council (ERC) under the European Union's Horizon 2020 research and innovation programme (Grant agreement No. 866411). The funders had no role in study design, data collection and analysis, decision to publish, or preparation of the manuscript.

**Competing interests:** The authors have declared that no competing interests exist.

**Abbreviations:** ABC, approximate Bayesian computation; BrdU, 5-bromo-2′-deoxyuridine; CDK, cyclin-dependent kinase; CI, confidence interval; DLS, double-labeled S-phase; EdU, 5-Ethynyl-2′-deoxyuridine; EGFP, enhanced green fluorescent protein; GFP, green fluorescent protein; NSC, neural stem cell; SCIP, Single Cell Identification Pipeline; SD, standard deviation.

stem cells make the decision to remain quiescent or enter the cell cycle individually and collectively.

Several determinants have been proposed as regulators of quiescence or division of stem and progenitor cells. Feedback mechanisms between stem cells and the environment have been found in several systems: In the adult mouse forebrain, neuronal activity in mossy cells or in granule cells regulates the activation of radial neural stem cells (NSCs) of the dentate gyrus [3,4]. In the subventricular zone, NSC cycling is inhibited by direct contacts with endothelial cells [5] or by the release of miR-204 from the choroid plexus [6]. In mouse and zebrafish adult neurogenic zones, Notch activity maintains NSCs in quiescence [7–9], in particular in the immediate neighborhood of dividing cells [7]. In other adult stem cell populations such as the epidermis, cell divisions are instructed by neighboring differentiating progeny [10,11]. Besides signals from the environment, cell-intrinsic modulations have been shown to impact on the proliferative activity, for instance, the metabolic control of lipogenesis that can induce proliferation [12,13]. Conversely, the expression of miR-9 [14] and the degradation of Ascl1 via the ubiquitin ligase Huwe1 [15] are factors inducing quiescence. While a combination of signals received from the environment and intrinsic to the cells seems to influence proliferating activity, it remains to be precisely understood whether and how cells coordinate their activity with their neighbors.

Toward this end, we examined the distribution of cells in S-phase in the pallial (dorsal) neurogenic niche of the adult zebrafish telencephalon, which is located on the outer surface of the brain [16,17]. The adult brain of the zebrafish and the telencephalon in particular is growing steadily [18] but at a slower pace with increasing age [19]. Radial glia constitute the entire ventricular surface of the brain with a single layer of cell somata, extending filopodial extensions tangentially [20] and long ramified processes through the parenchyme. They express—among others—Notch ligands, the transcription factors Her4 and Fezf2, the fatty acid binding protein BLBP, the enzyme Aromatase B, the intermediate filaments GFAP and vimentin [21–25] and a small percentage of them is dividing at any time point of observation [7]. The behavior of radial glia in steady state or injury conditions in this area of the brain hints to their function as NSCs giving rise to intermediate dividing progenitors and directly to neurons [23,26–28]. We previously found that cell cycle entries in NSCs occur with aggregated spatial patterns [29]. Here, we show that NSCs may undergo successive S-phases with short time windows, and modeling their activity creates similar spatiotemporal patterns as the ones observed experimentally.

## Results

### Neural stem cells in S-phase reveal aggregated spatial patterns on the dorsal ventricular zone

To understand cell cycle regulation in a population of mainly quiescent stem cells, we studied the spatial distribution of cells in S-phase (see Box 1) in the intact dorsal telencephalon (pallium) in whole mount preparations. We used adult *gfap*:GFP transgenic zebrafish, where NSCs express enhanced green fluorescent protein (EGFP) [30]. In order to assess which subset of cells is concomitantly in S-phase at a specific time point, we marked S-phases by the incorporation of the thymidine analog EdU 1 h before fixation (Fig 1A) and detected them by staining in whole mount preparations, followed by confocal microscopy (3D reconstruction shown in Fig 1B). We identified the 3D coordinates of *gfap*:GFP+ NSCs (Fig 1C, 1D and 1D') and of EdU+ nuclei (Fig 1C' and 1D) automatically. To analyze the spatial pattern of S-phases, we used the coordinates of all *gfap*:GFP+ NSCs (Fig 1D') as a reference grid (see S1A Fig). Via manual inspection, we then discriminated between EdU+*gfap*:GFP+ cells, representing NSCs

---

### Box 1. Statistical analysis of spatiotemporal point patterns

When events can happen at any point in a 2D Euclidean space, a **spatial Poisson process** leads to a point pattern that is completely defined by a single parameter, the density of points. This **null model of complete spatial randomness** has been used for diverse analyses, from forest structures [35] over accessibility of pediatric care [36] to road accident prevention [37]. In our case, the space analyzed is not Euclidean, but **discrete**, since NSCs in S-phase (the "events") only appear where NSC are already present in the zebrafish hemisphere. The corresponding null model is thus not a spatial Poisson process, but represented by the distribution of randomly sampled NSCs. Accordingly, we call a pattern of NSC in S-phase **random** if the distances between the events are not significantly different from randomly sampled NSCs in S-phase. We evaluate this with an adapted version of a spatial statistics measure called **Ripley's K** [31]. **Nonrandom patterns** can be classified into 2 types: When events are closer to each other than expected from the random null model, we call the pattern **aggregated**. If events are further apart from each other, we call the pattern **dispersed**. A simulated random division pattern is displayed in S1B Fig, and a simulated dispersed division pattern on the same hemisphere is shown in S1D Fig. The corresponding observed aggregated division pattern (with the same number of events) is shown in Fig 1E. We follow in our notation and nomenclature the book from Baddeley et al. [38], which provides an overview and examples of diverse spatial statistics methods.

---

in S-phase, and EdU+*gfap*:GFP− cells, representing intermediate progenitors in S-phase [23] (Fig 1D" and 1E). We quantified the distribution of NSCs in S-phase (Fig 1F) with an adapted discrete variant of Ripley's K [31] that measures the number of neighboring NSCs in S-phase observed in a particular radius, accounting for a possibly nonhomogeneous distribution of nondividing NSCs and edge effects (see Methods). In other biological contexts, Ripley's K analysis has been applied to analyze single-molecule localization, reviewed in [32], or to analyze single cells behavior [33]. According to this spatial statistics measure, NSCs in S-phase reveal an aggregated pattern (Fig 1F), which is significantly deviating from a random (S1B and S1C Fig) and dispersed pattern (S1D and S1E Fig) for radii >50 μm. Aggregated patterns of NSCs in S-phase were reproducibly found in approximately 70% of all hemispheres analyzed (S1F Fig), suggesting that S-phase entry happens in a spatially nonrandom manner. In the remaining hemispheres (30%), the observed patterns are random (S1G Fig). Following distinctions in the division activity described previously in Dray and colleagues [34], we compared division proportions and spatial patterns in 3 different hemisphere domains (lateral, medial, and anterior) and found that these domains also show aggregated and random, but no dispersed patterns. Since separate domains contain in approximately 40% of all cases less than 10 NSCs in S-phase and patterns often span across domains, we refrain from analyzing them separately.

## Neural stem cells in S-phase reveal aggregated spatiotemporal patterns

To investigate whether the spatial patterns of S-phases are influenced by previous cell cycle activity, we made use of a second thymidine analog, 5-bromo-2′-deoxyuridine (BrdU), and observed consecutive S-phases taking place in vivo. The 2 labels BrdU and EdU were administered by intraperitoneal injections separated by a labeling interval Δt of 32 h (Fig 1G). Fish were humanely killed, their brain dissected and fixed 1 h after the second administration

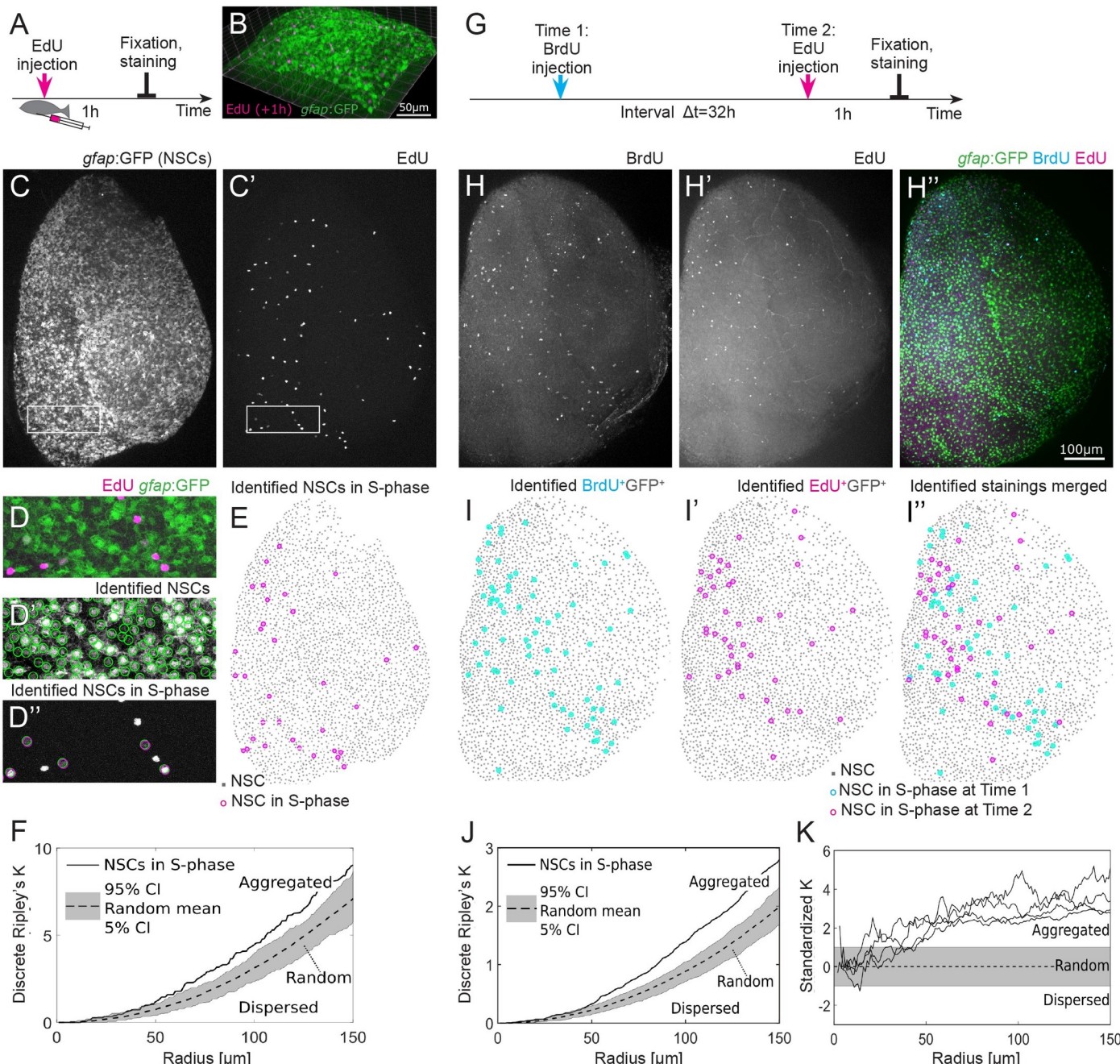

**Fig 1. S-phases occurring in the NSC population are spatiotemporally aggregated.** (A) Experimental setup: EdU is injected intraperitoneally 1 h prior to humanely killing the fish and fixation of the brain. (B) Part of the telencephalic hemisphere as a 3D reconstruction. The *gfap*:GFP transgene highlights cell bodies of NSCs, which are arranged on a 2D layer on the telencephalon surface, while their radial processes project deep in the parenchyme. (C, C') *gfap*:GFP in the whole hemisphere shown as a maximum intensity z-stack projection, anterior to the top, lateral to the right, and medial to the left. Boxed areas are shown as higher magnifications in (D–D"). (C') EdU coupled to Azyde-Alexa 647 highlights cells in S-phase and reveals their spatial distribution. (D) Merged GFP and EdU channel to identify specifically the NSCs in S-phase. (D') Automatically identified NSCs surrounded by green circles. (D") EdU+ cells were subdivided into EdU+*gfap*:GFP– and EdU +*gfap*:GFP+, the latter representing NSCs in S-phase. (E) The 33 NSCs in S-phase (pink circles) exhibit a nonrandom spatial pattern on top of all 2678 NSCs (gray dots). (F) Discrete Ripley's K quantification of the pattern shown in (E) reveals that NSCs in S-phase (solid line) are aggregated, i.e., closer to each other than expected from random (dotted line with 90% CI in gray) and dispersed patterns. (G) Cells in S-phase are labeled with BrdU and EdU with an interval of 32 h. Fish are humanely killed 1 h after the EdU injection, and the brains are imaged after fixation and staining. (H–H") Example of 1 telencephalic hemisphere, oriented as in (C), as a maximum intensity z-stack projection, in 3 different channels: BrdU (H), EdU (H'), and *gfap*:GFP transgene highlighting NSC bodies merged with the EdU and BrdU staining (H"). Scale bar: 100 μm. (I–I") Identified NSCs in S-phase at 2 different time points exhibit aggregated spatiotemporal patterns. (J) Discrete Ripley's K reveals more EdU+ NSCs around BrdU+ NSCs as expected from a random process. (K) We find spatiotemporally aggregated patterns with radii above 50 μm in all 4 hemispheres where S-phases have been labeled with a labeling interval of 32 h. BrdU, 5-bromo-2′-deoxyuridine; CI, confidence interval; EdU, 5-Ethynyl-2′-deoxyuridine; GFP, green fluorescent protein; NSC, neural stem cell.

(Fig 1G). EdU+ and BrdU+ cells were observed as clearly distinct sets (Fig 1H–1H"). When administered simultaneously as a control, EdU and BrdU reliably labeled the same set of cells (S1H Fig). We identified the 3D coordinates of the BrdU+ and EdU+ NSCs (Fig 1I–1I") and observed that some areas of the ventricular zone remain devoid of S-phase NSCs at those 2 time points. This observation was confirmed by a spatiotemporal Ripley's K analysis: From a radius >50 μm, the density of EdU+ NSCs around BrdU+ NSCs was higher than expected from random (Fig 1J). Thus, NSCs in S-phase aggregate also spatiotemporally, a trend that was observed in all 4 hemispheres with a labeling interval of 32 h (Fig 1K). These results suggest that subsequent S-phases are not randomly distributed and that the spatial organization observed is linked to the past activity in the population.

We then broadened the range of labeling intervals from 9 h to 72 h (S2A Fig). Using the BrdU–EdU double-labeling approach, we processed in total 36 hemispheres and identified NSCs and NSCs in S-phase at both time points (S1 Table). Visually, the observed patterns vary from homogeneous point clouds to strongly confined regions (Fig 2A). Quantitatively, we identify in 22 out of 36 hemispheres spatiotemporally aggregated patterns of divisions between 2 labeling time points, while the remaining 14 patterns are classified as random (see S2B Fig).

## An interaction model fits the observed spatiotemporal patterns

Ripley's K statistics is limited: It does not allow integrating different datasets, and it cannot quantitatively infer the strength and range of an observed pattern. To remedy these aspects, we use the temporal extension of a spatial model [29] that allows inferring the most likely parameters for interaction strength and interaction radius for an arbitrary number of datasets. Aggregated patterns emerge for an interaction strength >1, random patterns for strength = 1, and dispersed patterns for strength <1 (see Methods).

Applied to the 4 Δt = 32 h patterns shown in Fig 2A, we find that a model with an interaction radius of approximately 100 μm and an interaction strength >1 describes the data best (Fig 2B). On average, we find 202 ± 63 NSCs in total (mean ± standard devation (SD) from $n$ = 36 brains) and 6 ± 4 NSCs in S-phase in a 100-μm radius around an NSC in S-phase. Posterior sampling reveals a 90% confidence interval (CI) from 71 to 150 μm, a maximum likelihood interaction radius at 98 μm, and a 90% CI from 1.07 to 1.22 with a maximum likelihood interaction strength of 1.15. Applied to all 36 hemispheres with coordinates of 87,807 NSCs at distinct labeling intervals Δt (see S2C Fig), we find the most likely interaction radii to be around 100 μm (Fig 2C). The interaction strength (Fig 2D) is robustly above 1, indicating a significant ($p$-value = 0.0002 for a linear regression model) spatiotemporal aggregation of S-phase NSCs for all labeling intervals Δt.

## Division of NSC daughter cells reoccur within 24 to 72 h

Our analysis revealed a weak aggregation of successive NSCs in S-phase. Aggregated spatiotemporal patterns can emerge out of different mechanisms: Signaling waves that stimulate cells to divide in a particular region, activation via cell–cell contacts, or a behavior specified by the particular state of a cell. We thus inquired whether an NSC's history would be of relevance for new S-phase entries and made use of the double-labeling approach introduced above. Short labeling intervals up to Δt = 24 h revealed cells in S-phase incorporating both labels (Fig 3A–3F), hence denoted as double-labeled S-phases (DLSs). Remarkably, we also noticed cells with both S-phase markers that, unlike DLS, were arranged as doublets, i.e., 2 daughter cells close to each other. Such doublets allow us to extrapolate from the incorporation of S-phase labels to actual cell division events. One cell of these doublets entered a second S-phase, and these observations, denoted as redivisions, occurred with labeling intervals of Δt = 24 h, 32 h,

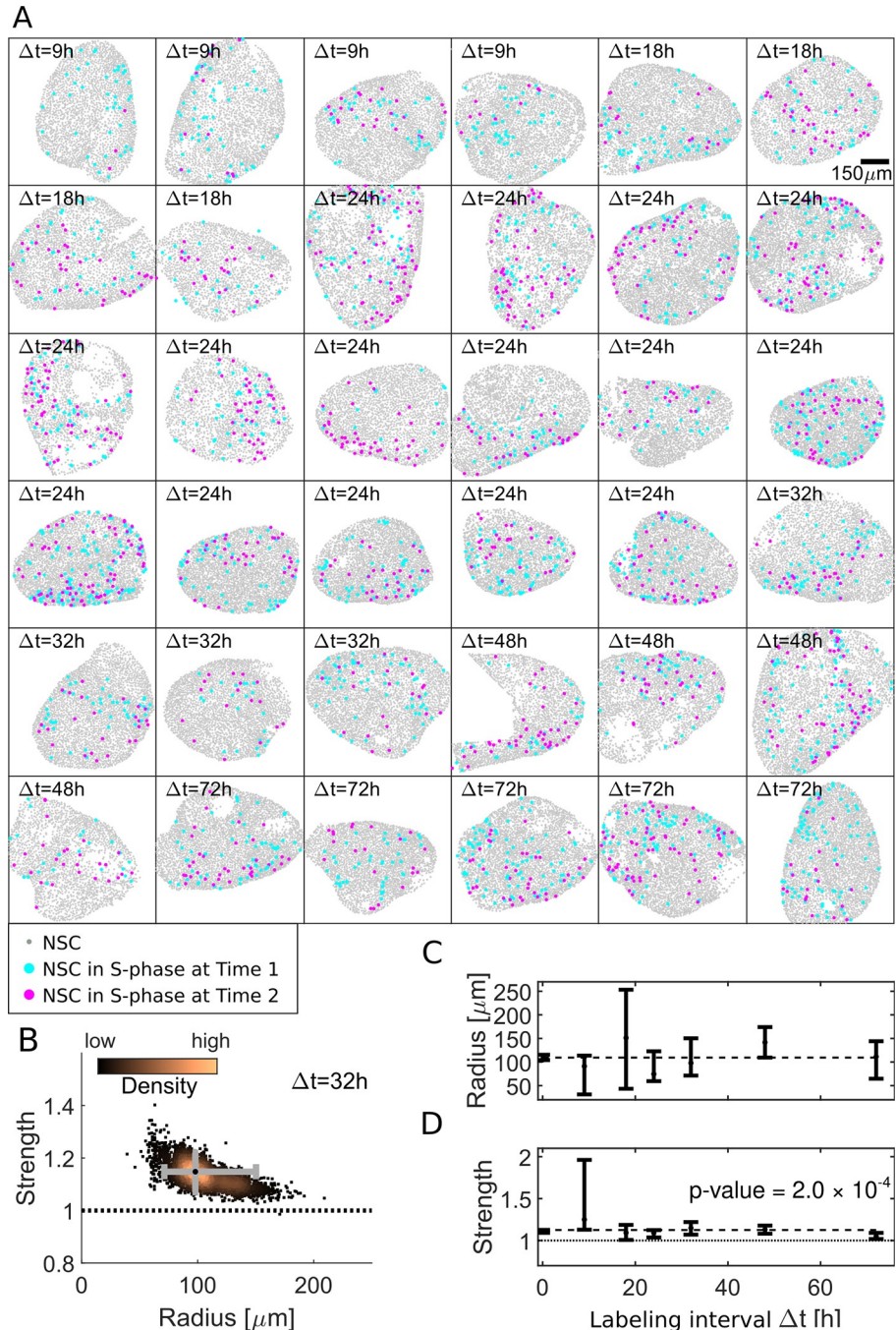

**Fig 2. Computational approach identifies an approximately 100-μm aggregation radius of NSCs in S-phase.** (A) Our dataset comprises 36 hemispheres with labeling intervals from Δt = 9 h to 72 h. (B) Posterior sampling identifies the most likely interaction strength of 1.15 and most likely interaction radius of 98 μm for 4 Δt = 32 h hemispheres. Whiskers (gray) cover the 95% CIs for strength and radius. Sampling point density is visualized from copper (high) to black (low). (C) Applied to all 36 hemispheres posterior sampling reveals an interaction radius around 100 μm. (D) The interaction strength is significantly above 1 for all labeling intervals Δt ($p$-value = 0.0002 for constant fit to most likely values) thus inducing aggregated patterns. CI, confidence interval; NSC, neural stem cell.

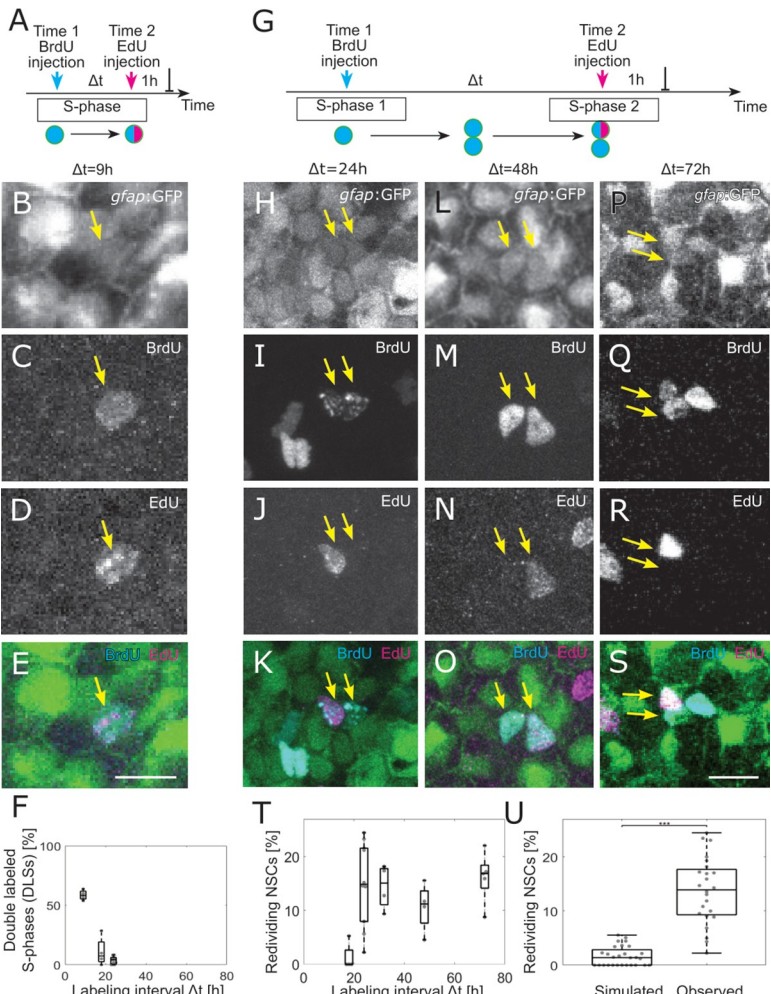

**Fig 3. A large proportion of NSCs redivide within 24 to 72 h.** (A) The 2 injections label the same cells in S-phase for small-labeling intervals, leading to NSCs that are both EdU and BrdU positive, denoted as double-labeled S-phase (DLS). (B–E) Example DLS (yellow arrow) for a labeling interval $\Delta t = 9$ h. (F) The DLS proportion is high for $\Delta t = 9$ h and decreases rapidly with increasing $\Delta t$. Each dot represents the value for 1 brain hemisphere. (G) After a division, 1 of the daughter cells already labeled by the Time 1 label can enter in a new S-phase and incorporate a second label. This cell thereby redivides. (H–S) Three examples of redividing NSCs with labeling intervals of 24 h (H–K), 48 h (L–O), and 72 h (P–S). Scale bar: 10 μm. (T) The proportions of redividing NSCs within the dividing NSCs at Time 1 remain high from $\Delta t = 24$ h to $\Delta t = 72$ h labeling intervals. Each dot represents 1 brain hemisphere. (U) Only 1.9±1.7% of randomly drawn divisions (same amount as observed per hemisphere) from all NSCs would be redrawn at random, while 14±8% of observed NSCs in S-phase reenter S-phase ($p = 9.4 \times 10^{-10}$, 2-sample Kolmogorov–Smirnov test). Box plots range from the 25th to the 75th percentile, and the central mark indicates the median and whiskers include points that are not more than 1.5 times the interquartile range away from the top or bottom of the box. BrdU, 5-bromo-2′-deoxyuridine; DLS, double-labeled S-phase; EdU, 5-Ethynyl-2′-deoxyuridine; GFP, green fluorescent protein; NSC, neural stem cell.

48 h, and 72 h (Fig 3G–3T). The observed redivision frequency of NSCs is significantly higher than expected from random: While $14 \pm 8\%$ of NSCs in S-phase reenter S-phase, only $1.9 \pm 1.7\%$ of randomly drawn divisions (same amount as observed per hemisphere) would be redrawn at random again ($p = 9.4 \times 10^{-10}$, 2-sample Kolmogorov–Smirnov test, Fig 3U). We are confident that we are observing this phenomenon in stem cells, as we could clearly distinguish between the events taking place in *gfap*:GFP+ and *gfap*:GFP− progenitors (S3A Fig). Reoccurring divisions in NSCs happened in similar proportions as in the *gfap*:GFP

– progenitors, the latter being considered so far as transit amplifying progenitors (S3B and S3C Fig and S1 Data, sheet S3C Fig). Notably, the *gfap*:GFP marker was the original label of the NSCs, without additional immunostaining for GFP, thus truly representing NSCs. Hence, S-phase NSCs display a high likelihood of undergoing another division within the following days.

To support these results, we performed independent experiments with a third time point of observation. We injected first BrdU, then EdU 24 h later, and dissected the fish another 24 h later, adding a PCNA immunostaining as a third cell cycle marker (S3B–S3Q Fig). In those brains, we found all combinations of NSC redivisions: BrdU+ daughter cells that incorporated EdU, BrdU+ daughter cells that expressed PCNA, and EdU+ daughter cells that expressed PCNA (S3K Fig and S1 Data, sheet S3K Fig for statistics). EdU+ doublets, which were also PCNA+ might represent cells that reached the end of a cell cycle without necessarily reentering a second division round. However, we also observed EdU+ doublets in which only 1 daughter cell was PCNA+, signifying a specific cell cycle reentry of this daughter cell.

We could not assess how many rounds of reoccurring divisions NSCs may undergo maximally, as an increasing number of BrdU-labeled cells with an increasing chase time renders a discrimination between neighboring clones impossible. However, experiments carried out with 72-h labeling interval revealed the presence of triplets of BrdU-labeled *gfap*:GFP+ cells (Fig 3Q), indicating that at least 2 rounds of divisions have been taking place within this time window. This implies that the distribution of cell cycle entries in the NSC population is also a result of the recent history of individual cells.

To assess whether cellular niches might be involved in the observed reoccurring divisions, e.g., via the formation of groups of NSCs with distinctive volumes, we segmented single NSCs in 3D whole mount brain images of 4 hemispheres and measure their volumes (see Methods and S3L–S3P Fig). We considered 3 groups of NSCs: (i) PCNA+ dividing NSCs; (ii) PCNA +BrdU+ and PCNA+EdU+ redividing NSCs; and (iii) nondividing NSCs without any marker. Measuring the volumes of all these NSCs and all immediately touching neighbors did not reveal any significant difference between the 3 groups (Kruskal–Wallis test, *p*-values = 0.1 and 0.37, respectively; S3Q Fig). Hence, the NSC volume does not reveal any distinctive organization around actively dividing NSCs, arguing against a model where cell density dependent niches would be associated with NSC activity.

## An agent-based model with redividing NSCs recreates aggregated spatiotemporal patterns observed experimentally

To quantitatively evaluate if redivisions suffice to induce the observed aggregated patterns, we simulated dividing NSCs with an agent-based spatiotemporal model. Such a model simulates the actions and interactions of autonomous agents, in our case NSCs. Here, every single cell can be modeled at every time point, while in other non-agent-based approaches, one only obtains summary statistics or averages per time point. However, fitting a spatiotemporal model to data is extremely challenging, since parameter estimation relies on repeated, computationally expensive simulations [39]. We thus split our simulation approach into 2 steps: First we fit a simple, nonspatial cell division model to the observed DLS and redividing NSCs for different labeling intervals Δt (Fig 3F and 3T) to derive the length and variability of cell cycle and S-phase. Second, we simulate an agent-based spatiotemporal model with the inferred parameters, analyze the simulated patterns statistically, and compare the results with experimental data.

To infer cell cycle kinetics from our data, we implemented a cell division model of dividing NSCs with 5 parameters (see Methods): the minimal cell cycle length $d_{cc}$ and minimal S-phase length $d_{sp}$, their variability $\beta_{cc}$ and $\beta_{sp}$ parametrizing a lag-exponential [40] distribution, and

the redivision probability $p_{rediv}$ (see S4A Fig). From this cell division model, we simulate dividing NSCs and a BrdU-EdU-labeling experiment (S4B Fig) and evaluate the percentage of redividing cells and DLS. We optimized the 5 parameters to the observed frequencies (see Methods) using approximate Bayesian computation (ABC) [41]. Our cell division model fitted the data, in particular the plateau of redividing cells (S4C Fig), and the sharp decrease of DLS cells after 9 h (S4D Fig). It estimated a minimal cell cycle time of 22.2 h with a mean cell cycle time of 107.5 h, a minimal S-phase length of 16.6 h with a mean S-phase length of 18.2, and a redivision probability of 0.38 (S4E and S4F Fig). Note that while the redivision probability is a parameter of our cell division model, the redivision fraction is an experimentally observable variable, which depends on the measurement method. Using snapshot measurements, we found a redivision fraction of 15%, which is considerably lower than the redivision probability of 38% (see S4B Fig for a detailed explanation).

We now fed an agent-based model with the inferred parameters to generate spatiotemporal patterns (see Methods and S4G and S4H Fig). To model cell shape kinetics, cellular potts models [42], vertex models [43], and cellular automata [44] have been used. We chose a cellular potts model, since it allows for arbitrary shapes and stochasticity in cell movement and is easily applicable as the default implementation in the Morpheus environment [45]. After a transient phase, we simulated a first measurement by marking all cells in S-phase at that time point. We simulated NSC kinetics further for different S-phase labeling intervals and simulated a second measurement, analogously to the first. We then created a distance matrix for S-phase cells at the 2 time points (exemplary patterns shown in Fig 4A for $\Delta t = 48$ h), calculated the discrete Ripley's K (see Fig 4B), and inferred interaction radius and strength, analogously to experimental data processing. Analogously to our experimental data, we found an interaction strength significantly $>1$ ($p$-value $= 6.6 \times 10^{-10}$) and an interaction radius around 45 μm (Fig 4C and 4D), which was a bit smaller than experimentally observed. We could also reproduce the observed heterogeneity in the emergence of the spatiotemporal patterns: In 36 simulated hemispheres, we found $58 \pm 8\%$ aggregated patterns ($n = 10$ independent simulations, mean ± SD) in accordance with our experimental observations (61%, see S2B Fig). Also in the simulations, the remaining patterns are classified as random throughout.

In control simulations, we omitted redivisions. In that case, all NSCs entered division with a rate of $1 \times 10^{-3}$ per hour (see Methods). In the corresponding simulated measurements (Fig 4E), neither the discrete Ripley's K (see Fig 4F) nor interaction parameters hint to an aggregation of divisions. While the interaction radius strongly varied around a mean value of $89 \pm 42$ μm (Fig 4G), the interaction strength is $0.97 \pm 0.2$, indicating random division patterns (Fig 4H). Thus, with redivisions and no additional assumptions, our agent-based simulations created aggregated spatiotemporal patterns of NSCs in S-phase as observed experimentally.

Finally, we analyze whether the agent-based simulations are able to reproduce the input parameters, i.e., observed DLS and redivision fractions, cell cycle length, and redivision probability. Redivision fractions and DLS fractions from simulations are similar to the ones observed in experimental data (S4I and S4J Fig), and cell cycle lengths observed in the simulations fit nicely to the probability density function of the lag-exponential distribution (S4K Fig). S-phase lengths are correct by default as we draw them after simulation for post-simulation analysis, from the respective lag-exponential distribution. The proportion of redividing cells in the simulation is confirmed with $38 \pm 0.5\%$ (S4L Fig).

## Discussion

In this study, we detected nonrandom, aggregated spatiotemporal patterns of successively dividing NSCs in the zebrafish brain using an experimental double-labeling approach and

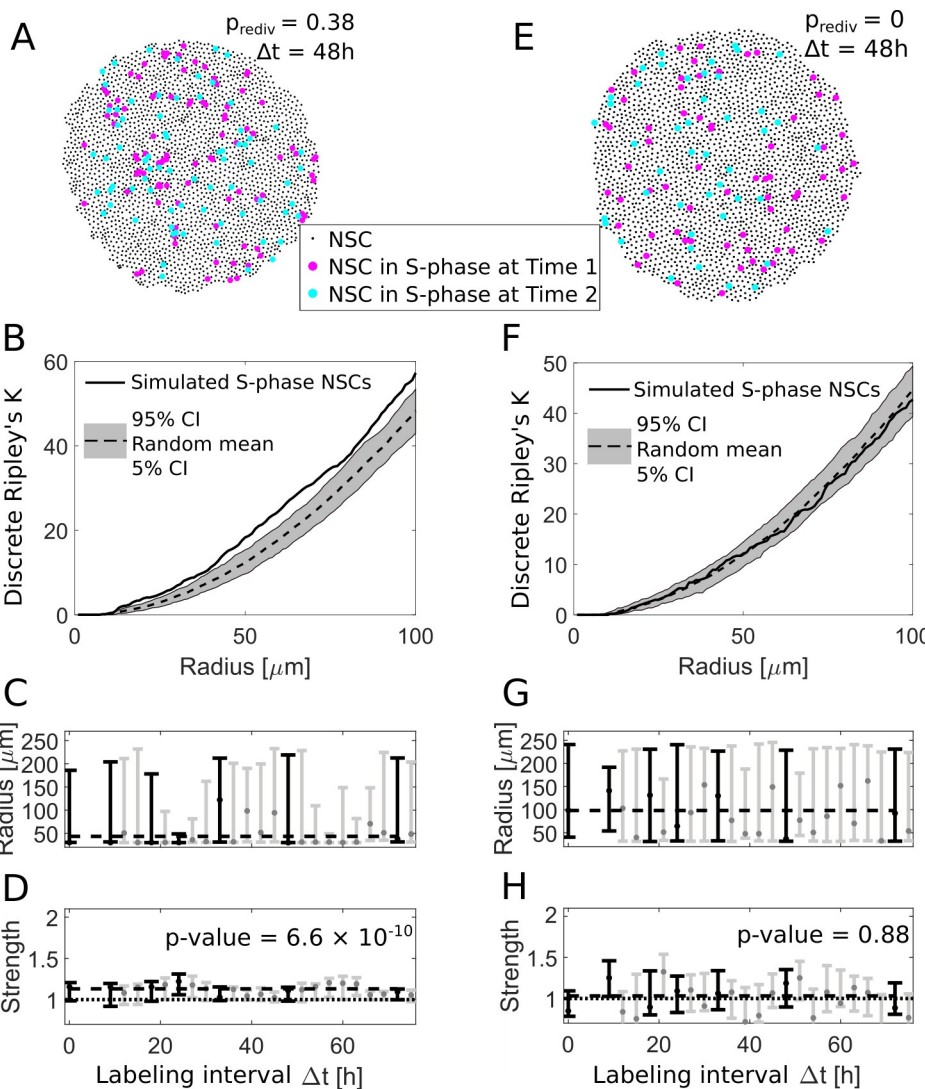

**Fig 4. An agent based redivision model can explain spatial aggregation of NSCs in S-phase.** (A) We use an agent-based model to simulate NSC divisions with a redivision probability of $p_{rediv}$ = 0.38 and perform virtual measurements with labeling intervals $\Delta t$ between 9 h and 72 h (here shown for $\Delta t$ = 48 h). (B) The simulated NSCs in S-phase in (A) exhibit an aggregated spatiotemporal pattern according to the discrete Ripley's K curve (solid line) which is above the 90% CI of randomly sampled patterns (gray area), similar to experimentally observed patterns. (C) The fitted radii for simulations with redividing NSCs for different labeling intervals $\Delta t$ are variable with a maximum likelihood value of 50 µm. Labeling intervals that are also available from experimental data (see Fig 2C) are shown in black, all others in gray. (D) The respective fitted strengths values are above 1 indicating aggregated patterns. Fitting a constant model to the most likely values with the same $\Delta t$ as experimentally observed (black bars) reveals a significant shift (*p*-value = $6.6 \times 10^{-10}$) from a strength of 1. Labeling intervals that are also available from experimental data (see Fig 2D) are shown in black, all others in gray. (E) Simulated NSC divisions and virtual measurements with a redivision probability $p_{rediv}$ = 0 at 2 labelings $\Delta t$ = 48 h apart. (F) Without redivisions, the simulated S-phase NSCs are within the boundaries of random patterns. (G) The fitted radii are highly variable with maximum likely values from 50 to 150 µm. Labeling intervals that are also available from experimental data (see Fig 2C) are shown in black, all others in gray. (H) In contrast to the simulations with redivisions, we now find no indication for aggregated patterns for all labeling intervals (*p*-value = 0.88 for a constant model with nonzero shift from a strength of 1). Labeling intervals that are also available from experimental data (see Fig 2D) are shown in black, all others in gray. CI, confidence interval; NSC, neural stem cell.

extended methods from spatial statistics for quantification and parameter inference. Moreover, we found a prominent cell cycle reentry in daughter NSCs.

The quantification of patterns of NSCs in S-phase is challenged by their heterogeneity and the relatively weak signal: Only 70% of the 36 hemispheres show aggregated spatial patterns, the remaining 30% qualify as random, so we wondered how this heterogeneity comes about. Since we estimate NSCs to redivide with only 38% probability, we believe that de novo divisions of previously quiescent cells dilute the aggregation patterns arising from reoccurring divisions. Our agent-based simulations are able to reproduce aggregated patterns and their frequency, however, with a smaller interaction radius then observed. Contributions to this deviation might come from the abstracted morphology of the hemisphere and individual cell shapes (see S4G and S4H Fig), the discrepancy between a 2D model and 3D effects in the brain, or the previously reported underestimation of radii when using a Euclidean distance measure [29].

A cell division model allowed us to estimate cell cycle and S-phase length distributions from double-labeling data. It is interesting to consider that, similarly to variations in the G1 and G2 phases [46,47], S-phase length can be variable too. Lengthening of S-phase occurs during, e.g., development [48,49]. The work by Arai and colleagues [50] and Turrero García and colleagues [51] report distinct lengths of S-phases according to distinct types of neural progenitors in the developing neocortex in mouse and ferret, respectively. In the adult mouse subependymal zone [52], the *gfap*+ B-type cells displayed very rapid S-phases of about 4 to 5 h, which is below the values we measured here of about 18 h by identifying double-labeled nuclei (see Fig 3F). Since we still detected occasional double-labeled nuclei with 24-h intervals, but not later, we can state with confidence that these nuclei were already in S-phase as the first label was present (until up to 4 h after the injection) and finished their S-phase as the second labeled was administered. Hence, the group of NSCs defined by *gfap*:GFP+ in the zebrafish telencephalon undergoes collectively and comparatively long S-phases.

The emergence of complex patterns from simple rules has been analyzed extensively, e.g., for artificial systems [53], biology-inspired models [54], and biological phenomena [44]. Here, we first use an interaction model to determine the strength and the radius of the observed pattern of S-phase NSCs. This extends traditional Gibbs or Cox models [38] to the analysis of a particular multi-type spatial point pattern where NSCs in S-phase appear at discrete locations defined by the presence of all NSCs. However, fitting an interaction model to spatiotemporal data does not necessarily imply that cell–cell interactions are present. We thus model quiescent and NSCs in S-phase as agents in a continuous space with stochastic cell cycle kinetics to quantitatively compare 2 hypotheses: Are extrinsic effects like cell–cell interactions or signaling waves required to generate the observed aggregated spatiotemporal division patterns, or do redivisions of NSCs suffice? We find that agent-based simulations with redividing NSCs suffice to explain the spatiotemporal patterns observed in the zebrafish brain.

According to this model, coordination of cell cycle entries in the population can be explained by the internal synchronization due to the cells' history. Such a model is supported by interesting studies following cell families in cultured cell lines, detecting correlations of cell cycle parameters between siblings and cousin cells as a result of inherited factors but independent on their location [55,56]. This cell intrinsic–driven behavior is in contrast to cellular systems endowed with a clear environmental regulation, in particular in stem cell compartments constituted by a distinctive 3D architecture like in the hair follicle in the skin. There, the precise location of stem cells is associated with their cycling behavior and fate [57], indicating that the position within an environment impacts on the stem cells activity. In the neurogenic zone studied here, no distinctive organization of the tissue would so far hint toward morphological features specifically associated with an NSC's activity. Our analysis of cellular volumes in this study does not indicate that a distinctive density dependent niche would be in place. In this

context, it is also interesting to note that 2 intermingled types of progenitors with radial morphology in the killifish telencephalon reveal distinct proliferative behaviors [58] even if sharing the same environment, arguing for a small contribution of environmental effects on stem cell activities.

Nevertheless, we cannot reject the existence of cell extrinsic mechanisms that might contribute to the spatiotemporal patterns, such as local diffusive signaling activity in delimited groups of cells, a functional activity of extended cell–cell contacts (as has been observed in NSCs by Obermann and colleagues [20]) or the activity of the Notch signaling pathway [7]. As a result of either mechanisms, levels of molecular heterogeneity have been observed: For instance, variable levels of the Zinc finger protein Fezf2 regulate Notch activity levels and quiescence of NSCs [25]. Likewise, the expression of miR-9 involved in keeping quiescence upstream of Notch signaling is found only in a subset of quiescent cells [14] several days after a division. And in a recent study, a subpopulation of NSCs that expresses low levels of Elavl3 has been characterized as mostly nondividing cells in transit toward neuronal differentiation [59].

Repeated divisions of NSCs were not expected, given the low percentage of divisions in the whole NSC population. NSCs were able to enter successive rounds of cell division within a few days, the earliest starting 24 h after the previous S-phase. This contrasts with the early model of adult NSCs established in the mouse dentate gyrus and subependymal zone, where radial astrocytes are quiescent and can replenish the transient amplifying progenitor population after all fast dividing cells have been eradicated by an ARA-C treatment [60]. The latest characterizations by single cell sequencing differentiate between quiescent versus active radial astrocytes; however, these most probably represent alternating states, as also suggested by a continuum of transcriptional states [61,62]. The definition of an active radial astrocytes here cannot distinguish between a repetitive or sporadic division behavior. Recently, however, evidence for several rounds of divisions within a few days in NSCs has been reported in the mouse subependymal zone from clonal analysis data using the Troy-driven recombination [63]. Further, live imaging of Ascl-driven recombination in the dentate gyrus in vivo demonstrated the existence of several divisions of radial astrocytes in a short time window [64]. In zebrafish, 2 studies based on in vivo imaging of the whole brain have followed NSCs, 1 observing single-labeled NSCs and their behavior of division and differentiation [28], the other considering the whole dividing NSC population with the help of 2 transgenic lines, her4:RFP and mcm5:GFP, highlighting the NSCs and the cell cycle, respectively [34], over the course of 2 weeks. Rapidly redividing daughter NSCs, as observed by our double-labeling approach in living fish combined with confocal imaging that allows for a clear distinction between nuclei close to each other, have not been reported yet. It is likely that they have been missed due to the difficulty to resolve distinct sister cells with intravital imaging in previous studies.

How these reoccurring divisions of adult stem cells come about will be important to assess in the future. Single cell sequencing data, such as performed by Cosacak and colleagues [65], might help associating these events with specific molecular pathways. Studies on human epithelial cell lines have shown that mother cells can relay distinct levels of CCND1 mRNA and p53 protein to their daughters, which after completing mitosis will then rapidly decide for the next round of division, depending on a resulting bimodal level of activity of cyclin-dependent kinase (CDK2) [66,67]. It could well be that in NSCs too, specific cell cycle regulators are transmitted to daughter cells, thereby permitting for immediate new rounds of division. Other mechanisms of daughter cells' decisions for quiescence or new cycle have been reported, for instance, in the adult mouse dentate gyrus, where degradation of Ascl1a in daughter cells mediated by the ubiquitin ligase Huwe1 takes place and promotes a reentry in quiescence [15].

Hence, in stem cell populations, individual phases of quiescence and exit thereof might well be predictable according to inherited factors, allowing us to understand how the kinetics of

tissue maintenance are regulated. Beyond this study, which aimed at understanding a regulation within days-scale time window, long-term tracing studies will help understanding the consequences of those patterns on the organization of the resulting neuronal circuits.

## Methods

### Zebrafish maintenance and transgenic lines

Zebrafish of the transgenic line *gfap*:GFP (mi2001) [30] were bred and maintained in the fish facility of the Helmholtz Zentrum München. Experiments were conducted under the animal protocol 55.2-1-2532-83-14, in accordance with animal welfare rules of the government of Oberbayern.

### Labeling of consecutive S-phases in vivo and immunostainings

BrdU or EdU were dissolved at a concentration of 1 mg/ml in saline solution (0.07% NaCl) containing methylene blue and injected intraperitoneally (5 ul/ 0.1 g body mass) into the fish at precise time points. Fish were over-anesthetized in MS-222 or placed in ice water for euthanasia, decapitated, and the brains dissected and fixed in 4% PFA overnight. Whole mount brains were processed for Click chemistry with Azyde-Alexa-Fluor-647 to detect EdU, following the manufacturer's protocol (C-10269, Thermo Fisher Scientific, Darmstadt, Germany); see also [68]. For the subsequent BrdU immunoreaction, brains were treated with HCl 2M at 37˚C for 30 min, washed in sodium tetraborate buffer 0.1 M, pH8 and in PBS, and incubated with Mouse-anti BrdU (Phoenix-Flow Systems, San Diego, USA, PRB1-U) 1:800 overnight. Mouse or Rabbit anti-PCNA (clone PC-10, Santa Cruz, Santa Cruz, USA, sc-56; Abcam ab15497) was diluted 1:800 or 1:100, respectively. Following secondary antibodies were incubated for 1 h at room temperature at 1:1,000 concentration: Goat-anti-mouse Alexa-Fluor-555, Goat-anti-mouse Alexa-Fluor-405, Goat-anti-Rabbit-Pacific Blue, and Goat-anti-Rabbit Alexa-Fluor-555 (Thermo Fisher Scientific).

### Imaging of whole mount brains

Brains were mounted in vectashield (Vector Laboratories, Burlingame, USA) or in PBS with 40% glycerol between 2 coverslips separated by 8-layered parafilm spacers. Imaging was performed using a Leica SP5 confocal microscope (Wetzlar, Germany) with a 20× glycerol immersion objective at a resolution of 2048 × 2048 pixels, and for close up views with a 63× glycerol immersion objective with a resolution of 1024 × 1024 pixels.

### Identification of *gfap*:GFP+ cells and PCNA/EdU/BrdU-labeled cells

NSCs that are labeled by GFP in the transgenic *gfap*:GFP line were identified using the Single Cell Identification Pipeline (SCIP; [29]). In this pipeline, single cells are automatically identified from an image 3D stack, exploiting the fact that all NSCs are located on top of the hemisphere on a 2D surface. SCIP returns x, y, and z coordinates for all NSCs. Nuclei labeled by PCNA immunochemistry, BrdU immunochemistry, or by EdU-click chemistry were identified semiautomatically: Nuclei were first identified with SCIP and then visually verified 1 by 1 using Fiji to avoid false positives and ensure correct assignment to GFP-positive or negative cells, using the 2 channels in consecutive z-planes of the confocal stacks. Images are displayed as maximum intensity projections of z stacks, single planes, or 3D visualization using the Plugins Clear Volume [69] or 3D viewer [70].

In experiments with Δt >18 h, the majority of Time 1-labeled cells were found as doublets, i.e., a pair of small cells close to each other (see Fig 3I and 3M and S3D–S3J Fig), since the time

span allowed the mother cell to reach its mitotic (M) phase. For Δt = 72 h, even triplets occurred (see Fig 3Q). Doublets and triplets were handled as single-division events in the subsequent analysis. Some of the doublets incorporated the second dU label and were categorized only to Time 1 in the spatiotemporal analysis, in order to assess distances specifically between the distinct sets of cells undergoing S-phases at consecutive time points. The proportion of redividing NSCs was calculated by dividing the number of doublets (or triplets) containing at least 1 GFP+BrdU+EdU+ cell by the total number of GFP+BrdU+ doublets (or triplets).

## Three-dimensional segmentation of *gfap*:GFP+ cells and PCNA/EdU/BrdU-labeled cells

Four-channel confocal images of 4 whole mount hemispheres taken at the 63× objective (1024 × 1024 pixels) were used. Pixel classification of each single channel was performed in Ilastik [71] to discriminate cells from cell borders and background. The Ilastik autocontext pixel classification was used for BrdU and PCNA channels containing background staining of blood vessels. The Fiji MorphoLibJ package [72] was then used on the pixel-classified GFP channel to perform either a 3D distance-transformed watershed segmentation or a marker transformed watershed segmentation using a white top hat filtered image as marker input, depending on the image quality and the precision of the resulting labeled image. Single objects in the segmented images were evaluated and quantified using the Fiji 3D Manager [73], and 3D reconstructions were obtained with the 3D Viewer [70]. Small objects with a volume below 30 $\mu m^3$, corresponding to segmented radial processes detached from segmented cell bodies, were filtered out from the analysis, giving a total number of 834, 961, 293, and 283 NSCs, respectively. Co-localization of segmented NSCs with the division markers was performed by quantification in the 3D Manager on binarized images and verified manually. The analysis of volumes of dividing, redividing, and quiescent NSCs, as well as their direct neighbors in the 4 hemispheres was performed in R using the spatstat package [38].

## Spatial statistics

Spatial statistics were applied on NSCs labeled in S-phase, i.e., *gfap*:GFP+ cells with EdU or BrdU staining. The set of NSCs without EdU or BrdU staining served as the substrate to, e.g., evaluate patterns for randomly dividing NSCs. Spatial analysis is performed with MATLAB (MATLAB version 9.7.0.1296695 (R2019b) Update 4).

## Three-dimensional distance matrix

We determine the distance between any 2 cells by calculating the shortest path on the hemisphere manifold to account for the bending of the hemisphere surface. To this end, we used the fitted hemisphere surface and calculated the stepwise shortest paths between 2 cell locations on it (see [29] for a detailed description).

## Ripley's K statistics

Ripley's K [31] is a measure for the deviation of a point pattern from spatial homogeneity. It has been previously applied in geographic information science [74], in the context of spatial economic analysis [75], archeological studies [76], single-molecule localization [32], or single cells behavior [33]. In all those applications, it is assumed that the point pattern occurs from a Poisson point process on a homogenous space. Here, however, division events can only occur where NSCs are located on the 2D hemisphere manifold. As the underlying NSC distribution could be inhomogeneous, we adapted the measure to account for this possible inhomogeneity

by sampling a point pattern only from discrete NSC locations and thus call it discrete Ripley's K.

For $S$ S-phase NSCs we calculate the spatial $K_S$ value for increasing radii $r$ along

$$K_S(r) = \left(\frac{S}{A}\right)^{-1} \sum_{i=1}^{S} w(i,r) \sum_{j=1,j\neq i}^{S} \frac{I(dist(i,j) \leq r)}{S} \tag{1}$$

This function counts NSCs in S-phase within a radius $r$ around the S-phase NSC $i$. The indicator function $I$ is 1 if cell $i$ and $j$ are closer than $r$ and 0 otherwise. The term is normalized by the total amount of NSCs in S-phase $S$ and by the S-phase NSC density $S/A$. The hemisphere area $A$ is calculated as the sum of all triangle areas between all NSCs obtained via Delaunay triangulation (see S1A Fig) [77]. The edge correction term $w(i,r)$ is 1 if the disc with radius $r$ around NSC $i$ does not cut the hemisphere edge and else calculated as the fraction of disc inside the hemisphere.

To obtain the spatiotemporal $K_{ST}$ between 2 sets of NSCs in S-phase, labeled at Time 1 and Time 2 we modified Eq 1:

$$K_{ST}(r) = \left(\frac{S_1}{A}\right)^{-1} \sum_{i=1}^{S_1} w(i,r) \sum_{j=1}^{S_2} \frac{I(dist(i,j) \leq r)}{S_2} \tag{2}$$

Here, we count $S_2$ (NSCs in S-phase at Time 2) cells around $S_1$ (NSCs in S-phase at Time 1) cells within $r$. In contrast to Eq 1 the whole term is normalized by the density of $S_1$ cells while the indicator function is divided by the number of $S_2$ cells.

To compare observed K values to random spatial distributions, we sample the amount of observed S and $S_2$ cells, respectively, ($S_1$ cell locations are fixed) from all NSCs 20 times and calculate the random sampling discrete K value. To evaluate whether the observed K value differs from random sampling, we check if the observed K value is below the 5% quantile (which would indicate spatial dispersion) or above the 95% quantile (which would indicate spatial aggregation) of the 20 sampled discrete K values.

To make Ripley's K plots comparable between hemispheres, we standardized the results per hemisphere similar to a z-score, such that K values are 1 when they are on the 95% quantile and −1 when they are on the 5% quantile.

To classify S-phase patterns as aggregated, random, or dispersed, we calculated the mean z-score between 30 and 150 μm. If the mean z-score is above 1 we classify the pattern as aggregated, between −1 and 1 as random and below −1 as dispersed.

## Model-based analysis

### Interaction model

To determine the spatial extent and the nature of the temporal interaction of divisions at different time points, we extend a spatial interaction model [29] with 2 parameters: the interaction strength ($g$), where $g = 1$ means no dependencies, below 1 dispersion between the 2 populations, and above 1 aggregation. The second parameter the model fits is the interaction radius ($r$) in μm.

$$\log(L(g,r)) = \sum_{i=1}^{S_1} \log(g^{\sum_{j=1}^{S_2} I(dist(i,j)\leq r)}) / \sum_{i=1}^{S_1} \log(g^{\sum_{k=1}^{N} I(dist(i,k)\leq r)}) \tag{3}$$

For each hemisphere, we fit the parameters ($g,r$) of the model to locations of cells in S-phase of 2 time points to detect spatial dependencies of NSCs in S-phase at the later time point ($S_2$) regarding NSCs in S-phase at the earlier time point ($S_1$). The parameters are fitted with a log-

likelihood approach (Eq 3), where $S_x$ is the number of cells in S-phase at time point x, while $N$ is the whole NSC population. The indicator function $I$ is 1 if the distance between 2 cells ($i, j$ or $i, k$) is smaller or equal $r$. In the numerator term, we iterate over all $S_1$ cells and count the number of $S_2$ cells having a smaller or equal distance than r, while in the denominator, we normalize the equation by counting the not affected (non-S-phase) cells of N within $r$ around $S_1$ cells.

Eq 3 calculates the log likelihood for 1 hemisphere. To calculate the log likelihood across several hemispheres, every single likelihood per hemisphere is summed up and optimized in parallel to form a combined likelihood. We use the PESTO toolbox [78] to maximize the likelihood including uncertainty estimation via posterior sampling [79] as can be seen exemplarily in Fig 2B for hemispheres with Δt = 32 h and for all hemispheres in S2C Fig.

## Cell division model

The observed fractions of redividing NSCs and DLS (Fig 3F and 3T) suggest an upper limit for the S-phase length of 32 h (since we observe no DLSs for Δt ≥ 32 h, Fig 3F) and a lower limit for all other cell cycle phases of 9 h (since we observe no redividing cells at Δt = 9 h, Fig 3T). A simple model with a fixed cell cycle length of, e.g., 32 h + 9 h, is however not able to explain a plateauing of redividing NSCs for Δt ≥ 24 h. Instead of constant cell cycle and S-phase length, we thus assume them to be distributed as delayed exponential distributions:

$$f(x; \beta, d) = \begin{cases} 0 & x < d, \\ \dfrac{1}{\beta} e^{-\frac{1}{\beta}(x - d)} & x \geq d \end{cases}$$

where β is the scale parameter of the exponential distribution and d is the delay. Note that [40] use a different notation for the 2 parameters. Additionally, we assume a redivision probability $p_{rediv}$ (S4A Fig), which is lower bounded by the observed redivision frequency of 15%, since we are not able to observe all redivisions (S4B Fig) in our snapshot experiments with an EdU labeling of 1 h.

To infer these 5 parameters (scale parameter and delay of cell cycle, $d_{cc}$ and $\beta_{cc}$, and S-phase, $d_{sp}$ and $\beta_{sp}$, and redivision probability $p_{rediv}$) from (i) the observed redivision frequency; and (ii) the fraction of DLS NSCs (see sketches in S4C and S4D Fig) at every labeling interval Δt = 9 h, 18 h, 24 h, 32 h, 48 h, and 72 h, we apply ABC [41]. Given the 5 parameters ($d_{cc}, \beta_{cc}, d_{sp}, \beta_{sp}$, and $p_{rediv}$), we simulate the dynamics of 10,000 dividing cells over 400 to 500 h. Using a fixed endpoint (Time 2), we can determine Time 1 by subtracting the according labeling interval. With 2 virtual measurements we calculate the observed redivision and DLS frequency via counting all cells being in S-phase at Time 1 and determine the status of these Time 1 S-phase cells at Time 2 (see S4B Fig): If an S-phase of a cell is labeled by both virtual measurements, it is classified as DLS, and if an offspring cell is labeled in S-phase, the cell is classified as redividing cell. The simulated redivision frequency is then the number of redividing cells divided by the number of S-phase cells at Time 1, and the DLS fraction is the number of DLSs divided by the number of S-phase cells at Time 1, accordingly.

In order to fit the observed data, we define a distance function between the observed and simulated redivision and DLS fractions per labeling interval. To this end, we calculate the sum of differences between means and SDs, respectively:

$$distance_{ABC} = \sum_i (|\text{mean}(obs_i) - \text{mean}(sim_i)| + |\text{std}(obs_i) - \text{std}(sim_i)|)$$

where $i \in \{9\,h, 18\,h, 24\,h, 32\,h, 48\,h,$ and $72\,h\}$. We optimize this distance function employing ABC with 50 epochs, evaluating 500 particles per epoch (ABC parameters).

## Agent-based spatiotemporal simulation

To assess the contribution of redivision events to the emerging spatial patterns of NSCs in S-phase, we simulated an NSC population using a cellular Potts model as implemented in Morpheus [45]. Morpheus is a modeling and simulation environment where cells act and interact as agents in space and time. We approximate a base division rate $p_{div}$ from the average observed S-phase cell fraction ($1.9 \pm 0.7\%$, $n = 36$ hemispheres, mean $\pm$ SD) and S-phase length estimation (approximately 18 h): $p_{div} = 0.019\,/18\,h = 1 \times 10^{-3}$ divisions per hour. This base division rate suffices for control simulations, but for simulations with redivisions, we decreased $p_{div}$ slightly. To obtain a similar amount of observed divisions as in the control simulations, we fixed $p_{div} = 9 \times 10^{-4}$ divisions per hour to account for redivisions, which divide after 1 cell cycle independently of the base division rate with probability $p_{rediv} = 0.38$ (inferred from the cell division model above).

At the beginning of the simulation, every cell has to wait at least 1 cell cycle duration until it is available for a spontaneous division via $p_{div}$. This leads to a small bias in the beginning (roughly 1/30th of the whole simulation time), but after all cells are older than their cell cycle time, there should be no impact anymore. We estimate the rate of NSC differentiation from the proportion of doublets with 1 *gfap*:GFP+ and 1 *gfap*:GFP− cell. This proportion is roughly 10% in our data. *Gfap*:GFP− cells are simulated for 1 more cell cycle duration and are then excluded, mimicking differentiated cells transitioning away from the stem cell pool inside the brain [27]. Cell cycle length is inferred from the cell division model (see above) with delay $d_{cc}$ and $\beta_{cc}$ from the delayed exponential distribution. Other predefined parameters are minimum and maximum cell size, determined by measuring real NSC size and simulated via sigmoidal cell growth. Morpheus implements basic cell–cell interactions and kinetic assumptions [45]. Our simulation starts with 500 cells and runs until the colony size reaches the observed $2,356 \pm 460$ (mean $\pm$ SD) cells (see Fig 4A–4E). We analyze cells in S-phase at the simulated measurement time points and apply the same spatial statistics as for the real data.

## Figures

Plots were prepared in Matlab, R, and Excel. Figures were adjusted in Adobe Indesign and Inkscape.

## Supporting information

**S1 Fig.** (A) Delaunay triangulation on the identified NSCs to calculate discrete Ripley's K. (B) A synthetically generated random pattern of NSCs in S-phase. (C) Discrete Ripley's K correctly identifies the pattern in (B) to be within the 90% CI of randomly sampled patterns. (D) A synthetically generated dispersed pattern with an interaction radius of 100 μm. (E) Discrete Ripley's K correctly identifies the pattern in (D) below the random interval. (F) Standardized K shows S-phase NSC aggregation in 24 (thick lines) out of 36 hemispheres. Hemispheres that show a mean standardized K value higher than 1 between 30 and 150 μm are classified as aggregated (thick lines). Each trace represents the spatial distribution of S-phase NSCs in 1 hemisphere. (G) All hemispheres of the dataset. Gray dots represent NSCs magenta dots represent NSCs in S-phase. Aggregation patterns are classified according to the standardized K analysis in (F). (H) Cells in S-phase are equally labeled by EdU or BrdU when those are administered concomitantly. BrdU, 5-bromo-2′-deoxyuridine; CI, confidence interval; EdU,

5-Ethynyl-2′-deoxyuridine; NSC, neural stem cell.
(TIF)

**S2 Fig.** (A) Experimental setup for measuring spatiotemporal patterns. With different labeling intervals Δt between the 2 S-phase labelings, we systematically profile spatiotemporal effects. (B) Standardized K shows spatiotemporal aggregation of S-phase NSCs in 22 out of 36 hemispheres for different Δt, while the remaining 14 patterns are classified as random. (C) Parameter inference on all labeling intervals Δt. Most likely interaction strength and radius for all hemispheres per labeling interval determined via posterior sampling density. Whiskers cover the 95% CIs for strength and radius. CI, confidence interval; NSC, neural stem cell.
(TIF)

**S3 Fig.** (A) Example of reoccurring divisions taking place in *gfap*:GFP− progenitors (yellow arrow). (B) Experimental setup for observing NSCs in S-phase at 3 time points within 2 days. (C) Reoccurring divisions have been quantified in 6 hemispheres out of 4 brains (% of BrdU+-clones labeled with a second cell cycle label) and are observed both in NSCs (*gfap*:GFP+, green dots) and in *gfap*:GFP− progenitors (gray dots) after 1 day (BrdU+EdU+) or after 2 days (BrdU+PCNA+). Higher percentages of redivisions are observed compared to the experiments with a short pulse of EdU labeling at time 2 (injection 1 h before killing, see Fig 3), probably due to more S-phases entries labeled with a longer duration of EdU availability in this experimental setup. (D-J) Maximum intensity projection and close-up view of a 4-channel image depicting recurrent divisions marked with 3 cell cycle labelings at 3 time points: 2 days earlier by BrdU (E), 1 day earlier by EdU (F), and at the time of fixation by PCNA (G). (H) *gfap*:GFP + daughter cell pairs (green arrows) and *gfap*:GFP− daughter cell pairs (white arrows) with respectively 1 daughter entering a second round of division. (I,J) Overlays. Scale bar: 10 μm. (K) Quantification of reoccurring divisions in NSCs performed on 63× magnification images from 4 hemispheres. The average proportions of dividing NSCs labeled at distinct time points reveal that about 40% of all PCNA+ NSCs have been generated by a recent division (compare gray quadrants PCNA+EdU+ and BrdU+EdU+ to the yellow PCNA-only quadrant) (L) 3D segmentation and 3D visualization of all NSCs performed on the same image as shown in (D–I), representing a total of 834 NSCs. (M–P) Close-up view of the boxed area delineated in D and L depicting BrdU-labeled sister cells, 1 of which is labeled with PCNA (yellow arrows). (M) Maximum intensity projection. (N) Single z-plane. All NSCs are delimited by white lines obtained by a 3D watershed segmentation applied after Ilastik pixel classification of the original GFP channel. 3D segmentation of the BrdU and PCNA channels are represented in cyan and yellow, respectively. (O) Labeled NSCs visualized in 3D (top view). (P) 3D reconstruction of the BrdU-labeled sister cells, 1 of which is PCNA-labeled. (Q) Volumes of all segmented NSCs from the hemisphere shown in L. No significant difference was observed between the volumes of NSCs found in reoccurring division (PCNA+EdU+ and PCNA+BrdU+), division (PCNA only) or quiescent state (negative for all cell cycle labels) according to Kruskal–Wallis test ($p$-value = 0.10). No significant difference can be detected either between the mean volume of the direct neighbors of those groups, arguing against the formation of distinctive niche areas within the NSC population ($p$-value = 0.37). BrdU, 5-bromo-2′-deoxyuridine; EdU, 5-Ethynyl-2′-deoxyuridine; GFP, green fluorescent protein; NSC, neural stem cell.
(TIF)

**S4 Fig.** (A) We devise a cell division model to describe dividing NSCs with 5 parameters: the minimal cell cycle length $d_{cc}$ and minimal S-phase length $d_{sp}$ and their variability $\beta_{cc}$ and $\beta_{sp}$, respectively, parametrizing a lag-exponential distribution and the redivision probability $p_{rediv}$. Note that we do not have any information about the length of G2 and thus do neither model

nor visualize this phase in the graphs. (B) We use observed redivision proportions (see Fig 3T) and the percentage of DLS cells (see Fig 3F) to feed a division model (S-phases labeled in red) and take simulated measurements at T1 and T2 to infer the redivision probability $p_{re\text{-}div}$. We observe a difference between the model ($p_{rediv}$ = 0.38) and observed redivisions frequency (0.15): From 10 simulated cells dividing at T1, 4 cells (cell 1, 3, 5, and 10, shown in bold) redivide. Thus, 40% cell redivide, which is close to $p_{re\text{-}div}$. However, our measurement at T2 only picks up 2 cells (cells 3 and 5) corresponding to only 20%, in line with the observed redivision frequency. (C, D) The cell division model fits the proportion of redividing cells (C) and DLS cells (D) for different labeling intervals. (E) We find an extremely variable delayed exponential distributed cell cycle with a minimal length ($d_{cc}$) of 22.2 h and a mean ($d_{cc} + \beta_{cc}$) of 107.5 h. (F) The S-phase length is narrow with 16.6-h minimal length ($d_{sp}$) and $\beta_{sp}$ = 1.5h. (G) Snapshot of an agent-based simulation using the Morpheus software corresponding to Fig 4A (Time point 2) with NSCs (green), NSCs in S-phase (red), and dividing intermediate progenitor cells (black). (H) Boxed area in (G) shown in higher magnification to visualize simulated cell shapes. (I, J) Proportion of observed redividing and DLS cells for different labeling intervals in agent-based simulations vs. experimental data. (K) Cell cycle length measured in the agent-based simulations compared to the probability density function of the estimated delayed exponential function in (E). (L) Fraction of redividing cells measured in 20 agent-based simulations. DLS, double-labeled S-phase; NSC, neural stem cell.
(TIF)

**S1 Table. For each brain hemisphere (Experiment) with a particular labeling interval, we detail the number of NSCs, number of S-phases at labeling time 1 (T1) and labeling time 2 (T2), number of NSCs in S-phase at T1 and T2, number of redivisions, number of NSC redivisions, number of DLS, and number of DLS in NSCs.** DLS, double-labeled S-phase; NSC, neural stem cell.
(XLSX)

**S1 Data. Data tables to support S3C and S3K Fig.**
(XLSX)

## Acknowledgments

We thank Carolin Loos and Lisa Bast for their advice regarding parameter inference, Yannik Schälte for his support with ABC, Benjamin Ballnus for his insights in MCMC sampling, Elmar Spiegel and Hannah Busen for statistical consulting, and the rest of the Marr lab for helpful comments. Also, we thank Walter de Back for his help to set up Morpheus and Jovica Ninkovic for his valuable remarks. Furthermore, we thank Moritz Thomas for their notes on the manuscript and Elly Pachel for her help on code/analysis reproducibility. The experimental work was carried out in the lab of Hernán Lopez-Schier.

## Author Contributions

**Conceptualization:** Carsten Marr, Prisca Chapouton.

**Data curation:** Prisca Chapouton.

**Formal analysis:** Valerio Lupperger.

**Funding acquisition:** Carsten Marr.

**Methodology:** Carsten Marr, Prisca Chapouton.

**Software:** Valerio Lupperger.

**Supervision:** Carsten Marr, Prisca Chapouton.

**Validation:** Valerio Lupperger.

**Visualization:** Valerio Lupperger.

**Writing – original draft:** Valerio Lupperger, Carsten Marr, Prisca Chapouton.

**Writing – review & editing:** Valerio Lupperger, Carsten Marr, Prisca Chapouton.

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
