## [Editor Report · Decision Letter 0]

4 Mar 2020

Dear Dr Marr, 

Thank you for submitting your manuscript entitled "Emergence of aggregated spatio-temporal division

patterns from re-dividing neural stem cells" for consideration as a Research Article by PLOS Biology.

Your manuscript has now been evaluated by the PLOS Biology editorial staff as well as by an academic editor with relevant expertise and I am writing to let you know that we would like to send your submission out for external peer review as a Short Report. Please note that PLOS Biology Short Reports have a limit of four main figures. 

Please re-submit your manuscript within two working days, i.e. by Mar 06 2020 11:59PM.

Kind regards,

Di Jiang, PhD

Associate Editor

PLOS Biology

---

## [Decision Letter · Decision Letter 1]

24 Apr 2020

Dear Dr Marr,

Thank you very much for submitting your manuscript "Aggregated spatio-temporal division patterns emerge from reoccurring divisions of neural stem cells" for consideration as a Short Reports at PLOS Biology. Your manuscript has been evaluated by the PLOS Biology editors, an Academic Editor with relevant expertise, and by four independent reviewers.

In light of the reviews (below), we would welcome re-submission of a revised version that takes into account the reviewers' comments including reviewer 1's suggestion in term of discussing different brain territories and niche effects. You will also need to reduce the number of main figures to four without removing any data from the entire manuscript. We cannot make any decision about publication until we have seen the revised manuscript and your response to the reviewers' comments. Your revised manuscript is also likely to be sent for further evaluation by the reviewers.

We expect to receive your revised manuscript within 2 months. 

**IMPORTANT - SUBMITTING YOUR REVISION**

*Re-submission Checklist*

*Published Peer Review*

*PLOS Data Policy*

*Blot and Gel Data Policy*

Sincerely,

Di Jiang, PhD

Associate Editor

PLOS Biology

REVIEWS:

Reviewer #1: This manuscript analyses the distribution of dividing adult telencephalic neural stem cells in the adult zebrafish brain. They determine that the distribution is weakly clustered (non-random) and by analysing divisions at two time points they find the locations of the second divisions are significantly correlated to the location of the first division. Some stem cells go through two successive divisions in a relatively short time. The authors interpret their data as evidence that a stem cell's recent history of division influences the probability of that stem cell dividing again. The authors favour a cell intrinsic mechanism for this phenomenon rather than local niche environment, but there is no evidence either way. I feel this work is still at the descriptive stage, somewhat preliminary and doesn't yet offer much insight into the biology of the topic.

Some things I would like to know:

If you compare the distribution of divisions across all brains is there a pattern in their locations or is it completely random? From Figure 3 it looks like more divisions at the periphery of the territory they analyse? And relatedly, are any of the division clusters related to known regional territories within the telencephalon?

Why do the authors favour a cell intrinsic mechanism to explain the non-random distribution rather than local niche environment?

What do the authors think might be the advantage to the non-random distribution?

Reviewer #2: The manuscript by Lupperger et al. with the title: "Aggregated spatiotemporal division patterns emerge from reoccurring divisions of neural stem cells" describes the analysis and mathematical modeling of a comprehensive dataset on patterns of adult neural stem divisions using whole mounts of the zebrafish telencephalon. Most of the stem cells in the zebrafish telencephalon are quiescent. The divisions occur in aggregated, non random patterns. A significant proportion of daughter stem cells reenter the cell cycle. Modeling the cycling behavior suggests that this re-entry into S-phase is a key determinant of the observed patterns of stem cell divisions. The authors conclude that a cell's cycling history is an important parameter in the spatiotemporal aggregation of dividing stem cells regardless of possible feedback mechanisms in the population. All together, this work addresses the important issue ie that of the balance between proliferation and quiescence of stem cells. The strength of this study lies in the fact that it is quantitative at a large scale and uses simulation as a tool to reveal new insights. I am in favour of publishing this paper. There are a few problems that need to be solved before. To assure a broad readership, the authors should make a bigger effort to bridge the gap between the disciplines. For mostly traditionally trained cell biologist the explanation of approaches and models needs to be improved. 

Specifically: 

I find the introduction and the description of the experiments extremely dense.

What is for example the difference between dispersed and random and aggregated and non-random. Please provide graphical models in the supplement illustrating the principle differences between the different arrangements. 

Are these aggregates, the same as previously determined region of higher proliferation activity? 

I am not sure whether Fig. 1B should be included in the main body. In any case it needs, guidance to anatomical details. Add an arrow to indicate telencephalon for example. 

Page 10 When they refer to the frequency of aggregated division patterns they observe this only for 70% of the cases. How do the cases look where this aggregation is not evident. Why is this the case? Is this a reflection of the biology of the experimental assessment of the patterns. What are the quality standards. Manual verification is mentioned. But what is the result of this? 

How similar are the patterns of divisions between different animals? 

The patterns of division appear non-random in space ie some regions are more proliferating than others. I guess this is what you refer to as aggregated? It would thus not be a surprise that in these regions cells within a certain radius will reenter cell cycle more frequently than anywhere else. Moreover, the GFAP-gfp positive cell bodies do not appear to be equally distribute but rather follow the pattern of dividing cells. Is this not a trivial result - the more NSCs in a region the more frequent cell cycle entry in this region? 

"To investigate whether proliferative activity regulates new S-phase entries, we analyzed spatial division patterns at different time points." I fail to see the evidence for regulation. It may be correlation. But the specific layout of the niche itself may provide the necessary positive cues to trigger cell cycle re-entry. Thus, re-division would then not be caused by the cell intrinsic history of divisions. 

What does "A positive interaction model fits the observed spatio-temporal patterns" mean for the biology of stem cells. Does this mean that a dividing cell is influencing a second cell to divide positively in a 100 micron radius. How many stem cells are in this area? Can one obtain really evidence for interaction from these models and data? The conclusion from the agent based spatio-temporal model is different. I find this confusing.

"To confirm our observations,….." In this and the following paragraph on page 5, I lack the statistical analysis of the statements. These need to be included as above in the text. 

"To quantitatively evaluate if re-divisions suffice to induce the observed aggregated patterns, we simulated dividing NSCs with an agent based spatio-temporal model."

Please do make an effort and give an explanation what this approach is and what the benefits are for a agent based spatio-temporal model. 

How do the patterns of division correlate with single cell sequencing data. Di these data provide markers for the NSCs prone to divide again? 

Reviewer #3: 

The manuscript entitled "Aggregated spatio-temporal division patterns emerge from reoccurring divisions of neural stem cells" by Lupperger, Marr and Chapouton (PBIOLOGY-D-20-00516R1) address a problem of aggregation of cycling stem cells in the zebrafish telencephalon. The authors investigate if neural stem cells transiting through the cell cycle S-phase are coordinated in space and time within the population of the adult zebrafish telencephalon. By using a combination of the Ripley's test and a previously published mathematical model (Lupperger et al., 2018. Cytometry A. 93(3):314-322), the authors demonstrated that neural cells transiting through S-phase are non-random but weakly aggregated, both in space and time. By using dual labelling in S-phase, the authors inferred cell cycle and S-phase length of the telencephalon neural stem cells and observed a rapid cell cycle re-entry in about 15 % of the newly born stem cells. The authors developed a simple non-spatial probabilistic mathematical model which they fitted to their experimental data on the rapid cell cycle re-entry to then use their best-fitted parameter values in a more complex spatial agent-based model. The combined models predict that the presence of cell cycle re-entry reproduces the Ripley's test results of aggregation whilst its absence leads to random spatio-temporal distribution of cells transiting through S-phase. The authors concluded that "spatio-temporal aggregation of dividing stem cells can thus emerge from the cell's history, regardless of possible feedback mechanisms in the cell population".

The problem of how neural stem cells regulate cycling in the neural tissue is of great importance in developmental biology and its repercussions can be naturally allocated in neuroscience and other more applied areas of medicine. Hence, the problem is very interesting for a variety of readers of PLOS Biology and the authors addressed it by using innovative methods. The more original aspect of the study, in my opinion, is the use of Ripley's statistics, a statistical method well known in other areas of science, to determine spatial aggregation. Although the authors previously studied aggregation of neural stem cell division in the same tissue in a previous publication (Lupperger et al., 2018. Cytometry A. 93(3):314-322), they did not use this statistical method in the previous publication. As a matter of fact, they anticipated that they would use this method in the end of the discussion section of the previous paper. The next original aspect of the study is that the authors combined the popular BrdU/EdU dual labeling with Ripley's statistics to now calculate aggregation of cells marked with both labels. Since both labels are not only separated in space, but more importantly in time, this allowed them to evaluate spatio-temporal aggregation of cells. Thanks to these two aspects the authors highlight a non-trivial spatio-temporal pattern of cells transiting through S-phase. I believe that both, the methodology (novel in developmental biology, to my knowledge) and the result itself are more than interesting. 

Nevertheless, there are a few points regarding the modelling results and the conclusions from it that need to be addressed. 

Major points:

1. The title of the manuscript has to reflect the specific findings of the study. The authors did not demonstrate that the aggregated spatio-temporal division patterns emerge from reoccurring divisions of neural stem cells, as the title of the submitted manuscript stays. Instead, they i) elegantly demonstrated the existence of an aggregated spatio-temporal division pattern, ii) they showed that a modelling approach incorporating cells re-entering into the cell cycle is sufficient to reproduce the aggregated pattern, iii) their model predicts that in the absence of cell cycle re-entering the aggregated pattern vanishes and iv) the study was performed in the adult zebrafish telencephalon. Hence, the manuscript title should reflect these four facts. Regarding the term "divisions", please see my Major comment Number 4.

2. The interaction radius predicted from the agent-based model is about half of the experimentally determined. Why is that? Could it be due to the different geometry of the in silico tissue compared with the telencephalon hemispheres? The first one is 2D and circular whilst the second one is 3D and not precisely circular. To test this hypothesis the authors could test other geometries more similar to the experimentally tested.

3. In the second section of Results, the authors performed a spatio-temporal distribution Ripley's K analysis of the experiments in which the labelling interval was 32 hours. Then, they broadened the range of labelling intervals from 9 hours to 72 hours. They quantitatively identified 22 out 36 hemispheres spatio-temporal aggregated patterns of S-phase cells between the two labelling time points. What is the explanation of this heterogeneity? What happens with the other 14 hemispheres (nearly 40 %)? Could this heterogeneity be explained with the models?

4. Along the text, the authors describe cell divisions. But the authors do not actually measure cell divisions. They measure cells transiting through S-phase. Of course, these are cycling cells that eventually will divide. But this is not what they measured. Maybe they also measured cells in mitosis but I could not find in the manuscript any reference to that. The only cases in which the authors could state that the cells did divide is when the daughter cells entered in a second S-phase. Therefore, the authors should write 'cells transiting through S-phase' or 'cells in S-phase' or something of the sort instead of 'dividing cells'.

5. In the second paragraph of the discussion section, the authors describe what they learn with their modelling approach. They state that " [we] quantitatively compare two hypotheses: Do we need cell-cell interactions or niche effects to generate the observed aggregated spatio-temporal division patterns, or do re-divisions of stem cells suffice? We find that the simplest model, i.e. one with no feedback but active, re-dividing NSCs is able to explain the emergent patterns observed in the zebrafish brains.". I partially agree with this conclusion. I agree that they showed that their model, which does not account for feedback, or any other mechanism for that matter, is sufficient to reproduce their results. But I find two problems with the rest of the argument. 1) They did not quantitatively compare the performance of a model with feedbacks, or any other mechanism, with neither their model nor with the experimental data; hence, the sentence is not accurate. 2) To term their model the simplest one because it lacks feedbacks, or any other mechanism, is at least debatable. I find many possible definitions of 'simplicity' and not all of them coincide with the one stated in the paragraph. Of course I understand what the authors imply but I consider that the concept of "simplicity" in this context is to vague and should be avoided. Overall, my conclusion is that that this paragraph should be rewritten.

Minor points:

1. In section "A positive interaction model fits the observed spatio-temporal patterns", the "positive interaction" is not properly defined in the main text. The model characteristics are better described in the Methods section, subsection "Model based analysis", sub-sub section "Influence model" and more details can be obtained in the previous publication of the authors above mentioned. But, the nature of the interaction and what "positive" means should be properly described in the main text.

2. In Figure 4 U, the authors perform a two-sample Kolmogorov-Smirnov test comparing the neural stem cells that are in S-phase at a given time point with those re-entering S-phase. Nevertheless, a key assumption of the KS test is that the two samples are mutually independent. Unfortunately, the set of the second sample is included in the set of the first one. Hence, unless I miss something, they are not strictly independent.

3. The authors feed the spatial model with the parameter values obtained from the simpler non-spatial model fitted to the re-cycling experimental data. These parameters are the minimal cell cycle length d_cc, the minimal S-phase length d_sp, their variability beta_cc and beta_sp and the re-division probability p_re-div. Did the authors verify that they properly work on the spatial model? In other words, did they verify that minimal cell cycle length and the minimal S-phase in the spatial model are what they should be? If they imposed a probability of re-division of 0.38. Did they empirically verify that the observed frequency of re-divisions is about 38 %? And so on... If the authors did these controls, they should be in the supplementary information.

4. In the last paragraph of the Results section the authors described control simulations in which all the neural stem cells entered division with a probability of 2 x10(-3) per hour. First, how this value was obtained? Second, a probability by definition is unit-less, I guess what they are reporting is a rate. They should express it as such.

5. In the Figure 1 K, the discrete Ripley's curve displayed corresponds to the only one hemisphere observed in panel D and J, right? If so, this should be explicitly stated on the Figure legend.

Reviewer #4:

(See attachment)

---

## [Decision Letter · Decision Letter 2]

9 Oct 2020

Dear Dr Marr,

Thank you for submitting your revised Short Report entitled "Agent-based modeling reveals that reoccuring neural stem cell divisions in the adult zebrafish telencephalon are sufficient for the emergence of aggregated spatio-temporal patterns" for publication in PLOS Biology. I have now obtained advice from the original reviewers 1, 2, and 3 and have discussed their comments with the Academic Editor. 

Based on the reviews, we will probably accept this manuscript for publication, assuming that you will modify the manuscript to address the remaining points raised by reviewer 3. You will note that reviewer 1 is still unpersuaded that your work offers the significance for PLOS Biology. However, in agreement with the Academic Editor, we think your study offers enough insights for a Short Report. 

We would also like to suggest a change to your title: "Reoccuring neural stem cell divisions in the adult zebrafish telencephalon are sufficient for the emergence of aggregated spatio-temporal patterns"

We expect to receive your revised manuscript within two weeks. Your revisions should address the specific points made by each reviewer. In addition to the remaining revisions and before we will be able to formally accept your manuscript and consider it "in press", we also need to ensure that your article conforms to our guidelines. A member of our team will be in touch shortly with a set of requests. As we can't proceed until these requirements are met, your swift response will help prevent delays to publication.

- a cover letter that should detail your responses to any editorial requests, if applicable

*Copyediting*

*Published Peer Review History*

*Early Version*

Sincerely,

Gabriel Gasque, Ph.D.,

Senior Editor,

ggasque@plos.org,

PLOS Biology

Reviewer remarks:

Reviewer #1: The authors have worked hard to improve their manuscript and I think its clarity and level of discussion are much better. I'm still a bit hesitant to say this is now appropriate for this particular journal as I still feel the biological significance of weakly aggregated patterns of neural stem cell divisions is unknown. So at the moment this is a nice analysis of a phenomenon whose function is uncertain. That definitely lowers its general interest to me. Also as the authors admit in their responses to reviewer comments, they cannot rule out that the aggregated divisions could result from local niche/environmental signals, so their proposal that this is driven by the cells' history is only consistent with their analyses, but is not proven and other mechanisms are possible. This also reduces the impact of the work for me.

Reviewer #2: Sufficently imoroved. No further comments

Reviewer #3: I consider that the authors addressed all my previous points. After reading the new version of the manuscript I find it significantly improved. I am in favor of publishing the manuscript after the following points are being addressed:

1) On the page 7, last subsection of the Results section, the authors wrote "We now fed an agent-based model, implemented in the Morpheus toolbox". This sentence omits the modelling class that the model has, which is definitely more important than the software that the authors used to implement the model. Along the manuscript there is no reference to the fact that the authors use a Cellular Potts Model. This modelling class should be acknowledged in the text of the manuscript, together with an argument of why they favored this modelling class. In fact, they could have chosen Cellular Automata or Vertex model, among others. But CPM has some advantages that the authors used and some disadvantages that the authors circumvented. Hence, it is important to at least briefly elaborate on this.

2) In the Supplementary Figure 4 B, the authors show that cells divide immediately after S-phase. Nevertheless, this implies that G2-phase is nonexistent. Why is that? Did the authors have an estimation of the length of this phase and know that it is negligible compared to the others? If this is the case, this should be acknowledged in the manuscript.

3) The authors experimentally detected that the probability of re-division was ~.15. Yet, the fitting of the simple model after the ABC fitting returned a probability of ~ 0.38. What is the posterior distribution of this parameter? The authors mentioned in the methods section, subsection "Cell division model" that they used .15 as a lower boundary. Thus, the experimental value and the model estimate are not significantly different from each other? If the authors fix this parameter in 0.15 in the model and repeat the ABC fitting, would they obtain similar results? The authors mentioned in the legend of Supp. Fig. 4 B this difference, but I think this point should be acknowledged in the results section. The authors should at least discuss why there is this discrepancy. 

4) In Legend of Fig. 4 A, there is a grammar problem. Please re-write. Also, what is the meaning of grey and black in Fig. 4 C, D, G and H?

---

## [Editor Report · Decision Letter 3]

17 Nov 2020

Dear Dr Marr,

On behalf of my colleagues and the Academic Editor, Catarina C Homem, I am pleased to inform you that we will be delighted to publish your Short Reports in PLOS Biology. 

PRODUCTION PROCESS

Before publication you will see the copyedited word document (within 5 business days) and a PDF proof shortly after that. The copyeditor will be in touch shortly before sending you the copyedited Word document. We will make some revisions at copyediting stage to conform to our general style, and for clarification. When you receive this version you should check and revise it very carefully, including figures, tables, references, and supporting information, because corrections at the next stage (proofs) will be strictly limited to (1) errors in author names or affiliations, (2) errors of scientific fact that would cause misunderstandings to readers, and (3) printer's (introduced) errors. Please return the copyedited file within 2 business days in order to ensure timely delivery of the PDF proof. 

If you are likely to be away when either this document or the proof is sent, please ensure we have contact information of a second person, as we will need you to respond quickly at each point. Given the disruptions resulting from the ongoing COVID-19 pandemic, there may be delays in the production process. We apologise in advance for any inconvenience caused and will do our best to minimize impact as far as possible.

EARLY VERSION

PRESS 

Kind regards,

Vita Usova

Publication Assistant, 

PLOS Biology

on behalf of

Gabriel Gasque,

Senior Editor

PLOS Biology